# reguloGPT: Harnessing GPT for End-to-End Knowledge Graph Construction of Molecular Regulatory Pathways

Xidong Wu[†]
*Electrical and Computer Engineering*
*University of Pittsburgh*
Pittsburgh, PA, USA

Sumin Jo[†]
*Electrical and Computer Engineering*
*University of Pittsburgh*
Pittsburgh, PA, USA

Yiming Zeng
*Hillman Cancer Center*
*University of Pittsburgh Medical Center*
Pittsburgh, PA, USA

Arun Das
*Department of Medicine*
*University of Pittsburgh School of Medicine*
Pittsburgh, PA, USA

Ting-He Zhang
*Department of Medicine*
*University of Pittsburgh School of Medicine*
Pittsburgh, PA, USA

Parth Patel
*Electrical and Computer Engineering*
*The University of Texas at San Antonio*
San Antonio, TX, USA

Yuanjing Wei
*Language Technologies Institute*
*Carnegie Mellon University*
Pittsburgh, PA, USA

Lei Li
*Language Technologies Institute*
*Carnegie Mellon University*
Pittsburgh, PA, USA

Shou-Jiang Gao
*Department of Microbiology and Molecular Genetics*
*University of Pittsburgh School of Medicine*
Pittsburgh, PA, USA

Jianqiu Zhang
*Electrical and Computer Engineering*
*The University of Texas at San Antonio*
San Antonio, TX, USA

Dexter Pratt
*Department of Medicine*
*University of California San Diego*
La Jolla, CA, USA

Yu-Chiao Chiu[‡]
*Department of Medicine*
*University of Pittsburgh School of Medicine*
Pittsburgh, PA, USA

Yufei Huang[‡]
*Department of Medicine*
*University of Pittsburgh School of Medicine*
Pittsburgh, PA, USA

*Abstract*—Molecular Regulatory Pathways (MRPs) are key to understanding biological functions. Knowledge Graphs (KGs) help organize and analyze MRPs by structuring complex interactions. Current methods for extracting KGs from biomedical literature struggle with hierarchical relationships and context. Large Language Models (LLMs) like GPT-4 show promise in addressing these issues but remain underexplored for end-to-end KG construction. We present reguloGPT, a novel GPT-4 based in-context learning prompt designed for the end-to-end extraction of a regulatory graph and context from a sentence that describes regulatory interactions. reguloGPT employs a context-aware relational graph to capture MRPs' hierarchical structure and resolves semantic inconsistencies by embedding context directly within the relational edges. We created a benchmark dataset comprising four hundred annotated PubMed titles on $N^6$-methyladenosine ($m^6A$) regulations. Rigorous evalu-
ations of reguloGPT on the benchmark dataset showed marked improvements over existing algorithms and other LLMs. We further developed a novel G-Eval scheme, leveraging GPT-4 for annotation-free performance evaluation that demonstrated agreement with evaluations on the benchmark dataset. Lastly, we constructed $m^6A$-KG by applying reguloGPT to 1,396 $m^6A$-related titles and demonstrated its utility in elucidating $m^6A$'s regulatory mechanisms of cancer phenotypes across various cancers. These results underscore reguloGPT's potential for advancing biological knowledge extraction. All reguloGPT works including source code, benchmark datasets, and $m^6A$-KG are available at https://github.com/Huang-AI4Medicine-Lab/reguloGPT.

*Index Terms*—context-aware relation graph, GPT, in context learning, knowledge graph construction, molecular regulatory pathways, $m^6A$ mRNA methylation, reguloGPT

These authors contributed equally to this work[†]. These authors are co-corresponding authors[‡].

This work is supported in part by funds from the National Institutes of Health (1SC3GM136594-02, 3R00CA248944-04S1, U01CA279618, U24CA269436, R00CA248944, R01CA124332, R01HG013359, R03OD036494, R35GM154967, P30CA047904, and P30DK120531) and the Leukemia Research Foundation.

## I. INTRODUCTION

Molecular Regulatory Pathways (MRPs) are central to our understanding of the molecular mechanisms controlling biological functions. Studying MRPs allows scientists to uncover disease-contributing dysregulations and guide the development of targeted therapies. For organizing and analyzing the ex-

tensive data within MRPs, Knowledge Graphs (KGs) have become instrumental. These KGs offer structured representations of complex interactions among various entities such as genes, proteins, and biological processes within MRPs [1], [2]. While databases like KEGG [3] have been established through meticulous human curation, the sheer volume and pace of new research publications pose a significant challenge to such manual efforts. To address this, automated Natural Language Processing (NLP) methods have been developed by combining rule-based and machine-learning strategies to improve the extraction of biomedical knowledge from literature, resulting in databases like INDRA [4].

MRPs are characterized by intricate relationships and hierarchical structures that span genes, proteins, and biological processes within a biological context. Text descriptions of MRPs, such as "METTL3-mediated m$^6$A methylation of SPHK2 promotes gastric cancer progression by targeting KLF2," suggest a context-specific relational graph involving several entities like METTL3, m$^6$A, SPHK2, KLF2, progression, and gastric cancer as context (Fig. 1A). The relational graph encompasses both explicitly and implicitly mentioned relationships that collectively define the mechanism by which METTL3 regulates the progression of gastric cancer.

Biomedical KGs are essential for integrating and analyzing complex data across various biomedical domains. KG can unify data from genomic databases, drug information, and research publications to provide a comprehensive view, enabling researchers to uncover insights that isolated data might miss and facilitating hypothesis generation by exploring connections between genes, proteins, diseases, and drugs, revealing new research opportunities and potential therapeutic targets. Additionally, KGs enhance the understanding of disease mechanisms and support context-specific predictions by comparing normal and disease-specific pathways, offering more precise insights for interventions. The advent of Large Language Models (LLMs) like GPT-4 represents a significant leap forward in Natural Language Processing (NLP), providing deep insights into the contextual dynamics of language. LLMs have transformed the traditional view of language from a static set of terms and rules into relational links between words [5]. This perspective aligns well with the core objective of KGs, which involves mapping out a network of relationships among entities. While LLM-based in-context learning (ICL) has demonstrated state-of-the-art performance in biomedical NLP tasks without expensive training or fine-tuning, its potential for end-to-end KG construction of MRPs remains largely unexplored, presenting a promising frontier in the field of biomedical research [5]. In this paper, we explore the capability of GPT-4 for the end-to-end construction of a context-aware relational graph to accurately delineate the context-specific MRPs of m$^6$A mRNA methylation from a given sentence. Our contributions are:

1) We proposed reguloGPT, a GPT-4 driven ICL prompt specifically designed for the end-to-end extraction of the regulatory graph and context to accurately interpret context-specific MRPs. We designed baseline, few-shot,

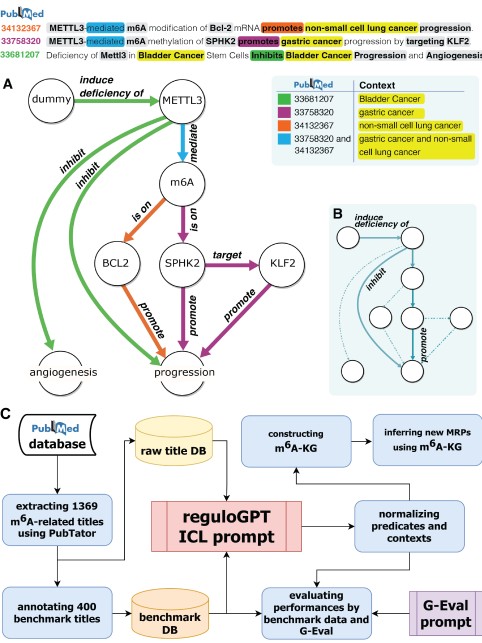

Fig. 1. (A) A context-aware relational graph proposed in reguloGPT. reguloGPT captures context-aware relational graphs from sentences (PubMed titles) depicting molecular regulatory pathways. The relational graph reflects the hierarchy of molecular pathways and incorporates the extracted biological contexts and associated PubMed IDs into edges. This context-aware edge enables the delineation of context-specific regulation. Alongside explicit edges, the graph includes implicit relationships, such as the link between KLF2 and gastric cancer, indicating KLF2's role in promoting the disease (extracted from the second title on top). (B) Excluding context in Knowledge Graphs (KGs) can lead to contradictory relations or misinterpretations. For example, without context, 'deficiency of METTL3' could be interpreted as either 'inhibit' or 'promote' the 'progression'. However, including specific contexts (e.g., bladder cancer vs. gastric cancer) resolves the contradictions as shown in (A). (C) The overall process of developing reguloGPT involves collecting data, creating a benchmark dataset, engineering ICL reguloGPT prompt, evaluating performance, generating a context-aware m$^6$A-KG, and utilizing m$^6$A-KG in downstream analysis.

and Chain-of-Thought (CoT) prompts for reguloGPT.

2) We introduced a context-aware relational graph representation of regulatory interactions in MRPs (Fig. 1A). This graph uniquely incorporates the context as part of the relational edges, thereby addressing and resolving the semantic contradictions of relations that often arise when contexts are not considered (Fig. 1B). It also possesses the inherent regulatory hierarchy of MRPs.

3) We annotated the context-aware relational graphs derived from 400 paper titles related to m$^6$A MRPs to create a benchmark dataset. This dataset encompasses a diverse array of contexts, entities, and relationships, making it highly valuable for the proper evaluation of reguloGPT.

4) We thoroughly evaluated the performance of reguloGPT for predicting contexts, entity recognition, and extracting both explicit and implicit relationships. reguloGPT demonstrated significant improvement over several existing algorithms and LLMs.

5) To overcome the need for manual annotation in evaluating reguloGPT, we introduced a novel G-Eval scheme, which leverages the CoT prompt to evaluate extracted context

and relational graphs. We demonstrated a strong similarity between G-Eval scores and annotation-based evaluations, suggesting its potential capability for annotation-free KG evaluation with larger datasets.

6) We applied reguloGPT to paper titles addressing m⁶A MRPs and constructed a comprehensive m⁶A-KG. We demonstrated its utility by exploring m⁶A-mediated pathways and delineated a mechanism that the m⁶A writer METTL3 regulates cancer-related phenotypes in breast cancer, lung cancer, and myeloid leukemia.

## II. Related Works

Several methods have been devised to extract relationships from text. These methods can be largely categorized as rule-based [6]–[9] and machine-learning-based [10]. However, they are mostly pipelined approach, focusing on extracting individual triplet only [11]. Extracting detailed graphs containing more than two entities from MRP descriptions challenges these triplet-based approaches, as they fail to consider correlations between triplets within a graph. This limitation results in cascading errors from misidentified entities, causing the extraction of redundant or missed relationships, and leads to exponential computational complexity [12]. Although a few recent works have considered graph extractions with three entities, such as drug-gene-mutation relationships [13], generalized approaches capable of extracting graphs of varying entity sizes are under-explored in the biomedical domain. Though the existing methods have their own advantage such as precision, control, interpretability, and adaptability, they are limited in capturing important contextual information like diseases and tissue types, potentially leading to inconsistencies and misinterpretations in biomedical KGs [14]. Recent advances in LLMs allowed researchers to address those limitations, providing a more integrated and adaptable approach to relationship extraction. Leveraging LLMs for knowledge graph construction provides advantages in handling complex and contextual relationships. Research and practical implementations are increasingly focusing on these capabilities to overcome the limitations of traditional triplet-based systems [15]–[18]. While LLMs have made noticeable strides in capturing multiple relationships, there are still challenges to address such as the effectiveness, scalability, and reliability of these methods in constructing detailed and accurate knowledge graphs [15]. Not only do these limitations in methods hinder efforts to address the challenges, but the lack of specialized benchmark datasets for large graphs of MRP also impedes progress. Existing datasets such as SemRep [6] have inspired new approaches for general biological relationships or chemical-disease relationships to some degree. However, they were not designed to represent the full spectrum of complex and context-dependent MRP graphs. Hence, there is an urgent need for annotated benchmark datasets and innovative approaches to extract these detailed, context-dependent relationships that are central to elucidating MRPs from the literature.

## III. Materials and Methods

### A. The reguloGPT workflow

reguloGPT is a GPT-4 prompt carefully designed and thoroughly assessed to extract a regulatory relational graph and its biological context from a sentence describing MRPs. The development of reguloGPT involves six key modules (Fig. 1C): ICL reguloGPT prompt engineering, data collection and annotation of the benchmark dataset, G-Eval prompt engineering, performance evaluation, context-aware m⁶A-KG construction, and downstream analysis of m⁶A-KG. We engineered three reguloGPT prompts (baseline, few-shot, and CoT prompt) to harness the full potential of GPT-4 for end-to-end context-aware relational graph extraction. To assess the efficacy of reguloGPT, we comprehensively evaluated its performance against the benchmark dataset and compared it with six existing algorithms. In addition to traditional evaluation, we further introduced an annotation-free evaluation scheme, G-Eval, and demonstrated its capability for future applications. We then applied reguloGPT to all titles to construct a context-specific m⁶A-KG and analyzed m⁶A-associated regulatory pathways, delineating context-aware relational graph representation and novel regulatory mechanisms across various cancers.

### B. In-context learning (ICL) prompts

To harness the potential of ICL, we developed three distinct reguloGPT prompts (Fig. 2A, B, and C). While the baseline prompt consists of instructions, definition, and output format (Fig. 2A), the few-shot prompt adds demonstrations (Fig. 2B) and the Chain-of-Thoughts (CoT) [19] prompt includes additional reasoning steps (Fig. 2C) with the demonstrations.

- *Instruction* specifies the task of reguloGPT for GPT-4
- *Definition* defines the nodes, edges, and context of the context-aware relational graph. Each edge includes two nodes and a predicate. We also incorporate implicit edges since many relationships are logically derived but not explicitly stated (Fig. 2A). Definition also outlines a set of constraints for extraction of nodes and edges.
- *Demonstration* provides a few examples containing pairs of sentences and the relational graphs extracted from the sentences, along with their corresponding contexts. We included four examples in our prompt (one is illustrated in Fig. 2B).
- *Chain-of-Thoughts* adds a series of intermediate reasoning steps for each example (Fig. 2C), encouraging a complex and logical response from LLM.
- *Output* provides instructions for the desired output format.

### C. Annotation of the benchmark dataset

The lack of benchmark datasets for context-aware MRP-related relational graphs is a primary obstacle to comprehensively assessing the proposed reguloGPT. To address this limitation, we developed an annotated benchmark dataset based on papers focusing on m⁶A mRNA methylation. m⁶A methylation, the most abundant mRNA modification in mammalian

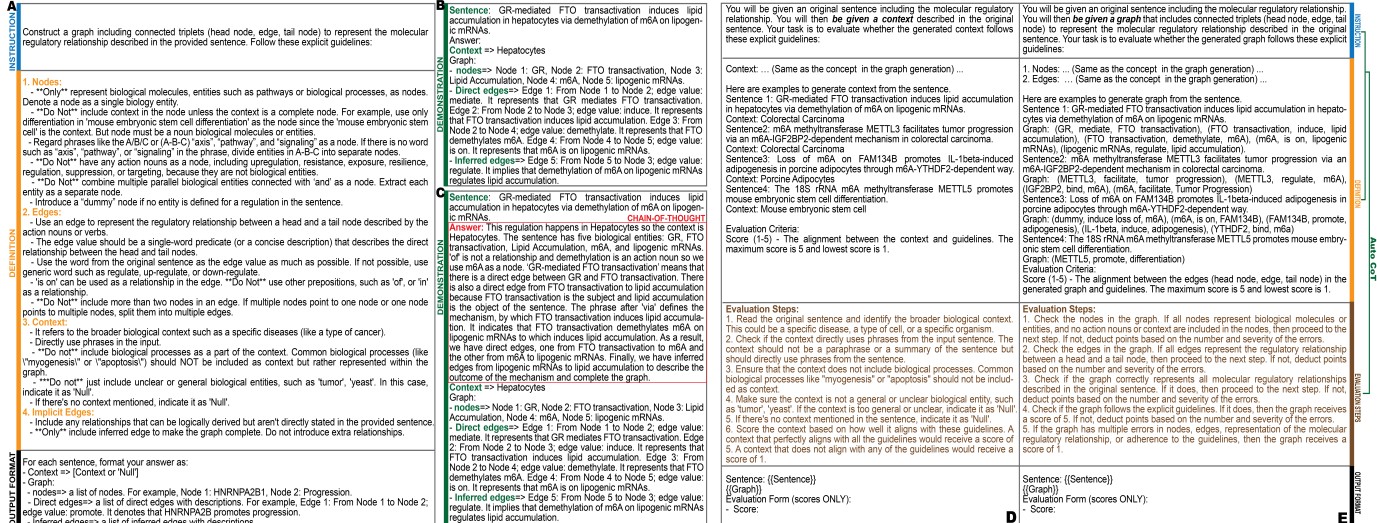

Fig. 2. reguloGPT prompts: (A) Baseline prompt including instruction, definition, and output format. (B) Demonstration in few-shot prompt. (C) Demonstration in CoT prompt. G-Eval prompt: (D) context evaluation and (E) graph evaluation. The Evaluation Steps were generated by GPT-4 based on our Instructions and Definitions. Then, they evaluate the context or graph added in the Output Format in a form-filling fashion.

cells, exhibits highly dynamic and complex mechanisms. $m^6A$ methylation has emerged as a highly active research area, which has garnered much interest in recent years. Creating the annotated dataset focusing on $m^6A$ circumvents the challenge of information overload inherent in more established research domains, yet effectively covers both general and nuanced MRP-related descriptions. We specifically focused on the titles of these papers as they provide the most concise description of molecular regulations. To compile this dataset, we searched PubMed using PubTator [20] with the keyword '$m^6A$' to extract papers published from 2013, the inception of this area, to 2023. Subsequently, we selected titles that encompass complete sentences with genes.To facilitate the annotation of a benchmark dataset, we assembled a team of five subject-matter-expert annotators with backgrounds in computer science and biomedicine to annotate 400 titles containing MRPs from the $m^6A$ research papers. The annotation process consisted of three steps: a) Practice phase: We randomly selected 20 sentences as practice examples. Annotators followed the instructions to identify nodes, edges, and context, engaging in discussions to reach a consensus. They also documented special cases for further annotation; b) Group annotation phase: Guided by the annotation guidelines established during the practice phase, we divided the 400 titles among the five annotators. Each annotator reviewed and annotated a subset of titles, after which they cross-reviewed another annotator's work c) Adjudication phase: Annotations were considered final if both annotators agreed on them. In cases of disagreement, the annotations were discussed within the group until a consensus was reached.

### D. Normalization of nodes, edges, and contexts

We used Gilda [21] to normalize nodes initially. Subsequently, we performed manual normalization to ensure consistency. Nodes were further categorized into five types: **$m^6A$,**

**$m^6A$-Writers/Erasers/Readers ($m^6A$-WER), Gene_Protein, GO_Pathway**, and **Other**. For edge normalization, we updated the ontological predicate definitions in SemRep [6], resulting in a dictionary of 32 predicate types: administered to, affects, associated with, augments, causes, coexists with, compared with, complicates, converts to, diagnoses, disrupts, higher than, inhibits, interacts with, isa, locates, lower than, maintains, manifests, methods, occurs in, part of, precedes, predisposes, prevents, process of, produces, same as, sensitizes, treats, and uses. Similarly, we normalized contexts adopting cancer types based on definitions from The Cancer Genome Atlas (TCGA) [22].

### E. Metrics and criteria for annotation-based evaluation

We adopted relaxed entity matching scheme used in [23] and evaluated the performance as follows: a) Node: A node is considered true positive if it achieves more than 50% similarity with the ground truth. Nodes failing to meet this criterion are considered as false positives. Any ground truth nodes not predicted are counted as false negatives; b) Edge: After normalizing using the predicate dictionary, each edge is evaluated based on its predicate and the node alignment. An edge is true positive if both the predicate and node alignment matched the ground truth. False positives occur when node alignment is correct, but the predicate does not match. Any ground truth edges not predicted were false negatives; c) Context: Accuracy serves as the metric for context prediction. It is considered correct if achieving more than 50% similarity with the ground truth context.

### F. G-Eval for annotation-free assessment of reguloGPT

Since manual annotation is labor-intensive and costly, assessing context-aware KG construction based on human annotations at scale is practically infeasible. Recent research on annotation-free evaluation of Natural Language Generation has leveraged LLMs directly as evaluators [24]. For

example, GPTScore utilized LLMs to evaluate candidate outputs, assigning scores based on generation probability without referencing any human annotations. Following a form-filling paradigm [25], GPT-4 was also able to assess the coherence, consistency, fluency, and relevance of generated texts, with high agreement with human evaluators. Inspired by these exciting studies, we proposed G-Eval (Fig. 2D and E), a novel annotation-free evaluator based on GPT-4 for reguloGPT output. G-Eval employs the form-filling paradigm, encompassing a context evaluation (Fig. 2D) that assigns a score to the extracted context, and a graph evaluation (Fig. 2E) that scores the final extracted graph from a sentence. The prompts for context and graph evaluations include four parts: 1) Introduction; 2) Definition, which describes the concept of context in the context evaluation, or the concepts of nodes and edges in the graph evaluation; 3) Evaluation Steps; and 4) Output Format. These concepts—contexts, nodes, and edges—are consistent with those defined in the reguloGPT prompts (Fig. 2A). The evaluation steps were generated by GPT-4 based on the introduction and definition. Scores range from 1 to 5, and G-Eval repeats the evaluation five times to obtain an average score as the final output [26].

### G. Construction of the $m^6A$ knowledge graph

In addition to the annotated benchmark dataset, we processed 969 $m^6A$-related titles with our reguloGPT-CoT prompt to extract context-aware graphs. We applied normalization to standardize the extracted nodes, edges, and contexts. These normalized relational graphs were integrated with those from our benchmark dataset by linking common nodes and edges to construct $m^6A$-KG, a comprehensive KG detailing $m^6A$ functions across diverse contexts. A distinctive feature of $m^6A$-KG is that each edge incorporates a set of associated contexts to inform the context under which the regulation defined by the edge occurs. Each edge also includes the extracted raw edge values and PubMed Identifier (PMID) of the associated titles, enabling traceability back to the original paper for reference. Neo4j was used [27] for visualizing and managing $m^6A$-KG.

## IV. RESULTS

### A. The benchmark dataset

Our query of PubMed yielded 1,369 research papers whose titles describe $m^6A$ MRPs. From these, we selected 400 titles and meticulously annotated their context-aware relational graphs addressing the regulatory mechanisms of $m^6A$ mRNA methylation to create the benchmark dataset. The benchmark data comprises 787 nodes, 1374 links with 153 predicates, and 243 contexts: equivalent to a graph with an average of 3.71 triplets and 0.86 context per title. Subsequent normalization resulted in 695 unique nodes, 1282 links with 21 predicates, and 160 unique contexts including 49 cancer contexts.

### B. reguloGPT outperforms existing algorithms and other LLMs

We evaluated reguloGPT's performance against manual annotation using the benchmark dataset, comparing it to four

TABLE I
NODE, EDGE, AND CONTEXT PREDICTION RESULTS*

| | Node | | | Edge | | | Context |
|---|---|---|---|---|---|---|---|
| | F1 | Re | Pr | F1 | Re | Pr | Accuracy |
| EIDOS | 0.74 | 0.60 | 0.98 | 0.37 | 0.26 | 0.68 | - |
| REACH | 0.69 | 0.55 | 0.94 | 0.22 | 0.15 | 0.42 | - |
| TRIPS | 0.55 | 0.47 | 0.66 | 0.12 | 0.08 | 0.20 | - |
| OpenIE | 0.63 | 0.69 | 0.58 | 0.18 | 0.19 | 0.17 | - |
| Mixtral-CoT | 0.95± 0.001 | 0.94± 0.002 | 0.96± 0.002 | 0.56± 0.004 | 0.56± 0.003 | 0.56± 0.005 | 0.79 |
| Llama3-CoT | 0.93± 0.000 | 0.95± 0.001 | 0.91± 0.001 | 0.51± 0.001 | 0.56± 0.002 | 0.46± 0.001 | 0.73 |
| reguloGPT-baseline | 0.95± 0.002 | 0.94± 0.002 | 0.96± 0.002 | 0.56± 0.006 | 0.50± 0.006 | 0.65± 0.007 | 0.74 |
| reguloGPT-fewshot | 0.95± 0.002 | 0.92± 0.003 | 0.98± 0.002 | 0.58± 0.003 | 0.51± 0.003 | 0.67± 0.005 | 0.87 |
| reguloGPT-CoT | 0.96± 0.001 | 0.94± 0.002 | 0.97± 0.001 | 0.60± 0.005 | 0.55± 0.005 | 0.66± 0.005 | 0.89 |

* F = F1 score, Re = recall, Pr = precision

TABLE II
G-EVAL RESULTS

| | Context | | Graph | |
|---|---|---|---|---|
| | score* | similarity† | score | similarity |
| Baseline | 3.74 | 0.81 | 3.76 | 0.61 |
| Few-shot | 4.19 | 0.84 | 4.59 | 0.78 |
| CoT | 4.35 | 0.84 | 4.67 | 0.81 |

* The range of scores is 1 - 5.
† The similarity denotes the rand similarity coefficient between the G-Eval and human annotation evaluations of reguloGPT's prediction on the benchmark dataset at the sentence level.

established algorithms as baselines: OpenIE [8], TRIPS [9], REACH [7], EIDOS [10]. Note that these algorithms do not output context information, so context evaluation results of these algorithms are not presented. Besides these baseline algorithms, we also evaluated the CoT prompt with Mixtral-8x22B and Llama3-70B, two widely used open-source LLMs. Mixtral-8x22B is the latest and largest mixture-of-experts LLM from Mistral AI, known for its efficient architecture and strong performance. Llama3-70B is the most advanced model in Meta's Llama series, featuring 70 billion parameters and optimized for instruction-following tasks. These models represent state-of-the-art performance in publicly available language models and provide robust comparisons against reguloGPT. Overall, ICL prompts powered by LLMs demonstrated remarkable superiority over non-LLM algorithms (Table I). reguloGPT-CoT exhibited the most effective performance, achieving an impressive accuracy of 0.89 for context prediction and F1 scores of 0.96 for node prediction and 0.60 for edge prediction. The relatively lower performance on edge prediction underscores the inherent complexity in accurately extracting a graph. Compared to other LLMs, reguloGPT-CoT achieved higher F1 score than both Mixtral-CoT and Llama3-CoT, confirming the consensus on the superior ability of GPT-4 for text understanding and reasoning over other LLMs [28]. Though EIDOS showed the highest performance among baseline algorithms, reguloGPT-CoT achieved substantial improvements with 22% and 23% higher node and edge F1 scores than EIDOS, respectively. The marked increase in edge recall highlights the advantage of reguloGPT's end-to-end strategy

for graph extraction over methods focusing on individual triplets separately. The improvement is evident in the title "The m⁶A methyltransferase METTL3 promotes osteosarcoma progression by regulating the m⁶A level of LEF1." The benchmark annotations include four triplets under the context of "osteosarcoma," however, REACH only identified 'METTL3 – [*STIMULATES*] – level of LEF1.' Similarly, EIDOS extracted only one triplet 'm⁶A methyltransferase METTL3 – [*STIMULATES*] – osteosarcoma progression.' In contrast, all three reguloGPT ICL prompts successfully extracted all four relations with the correct context. Among the three reguloGPT ICL prompts, the advanced CoT prompt produced the most aligned output with our requirements. For instance, although we asked GPT-4 to introduce a dummy node to capture the state of an entity in the prompt, the baseline prompt ignored this guideline. In contrast, by adding one example in a demonstration with a similar case, the few-shot prompt was able to follow the requirement, even though this alignment was not stable. For instance, in the title "Suppression of m⁶A reader Ythdf2 promotes hematopoietic stem cell expansion," the few-shot prompt neglected the condition, but the CoT prompt successfully maintained this alignment.

### C. G-Eval is consistent with benchmark evaluation

We next investigated the G-Eval evaluations of the three reguloGPT prompts on the benchmark dataset and assessed the consistency between G-Eval evaluations and benchmark annotations. G-Eval produced scores for context evaluation and graph evaluations for each title. Examining the average score across the 400 titles (Table II) revealed a consistent trend with the annotation-based evaluation in Table I, i.e., the CoT prompt exhibited the best performance, followed by the few-shot and baseline prompts. To further validate the effectiveness of our G-Eval strategy, we analyzed the similarities between the G-Eval scores and annotation evaluation. Since the annotation evaluation for each sentence is binary, i.e., correct or incorrect, we first binarized G-Eval scores based on the score distribution, converting a score to correct if it was greater than 3 and incorrect otherwise. Furthermore, since G-Eval conducts graph evaluation instead of evaluating nodes and edges separately as the annotation-based evaluation does, we also generated a graph-level annotation evaluation where a sentence was deemed correct if more than 50% of the edges in the sentence were correctly predicted. We did not consider node prediction because their F1 scores are high, as shown in Table I. To compare the similarity between the G-Eval and annotation evaluations, we computed the Rand matching coefficient for each title. The similarity results demonstrate high similarities between the two evaluations, especially for CoT, where the Rand similarities reach 0.84 for context prediction and 0.81 for graph prediction. These results suggest that G-Eval is a promising annotation-free method for evaluating reguloGPT.

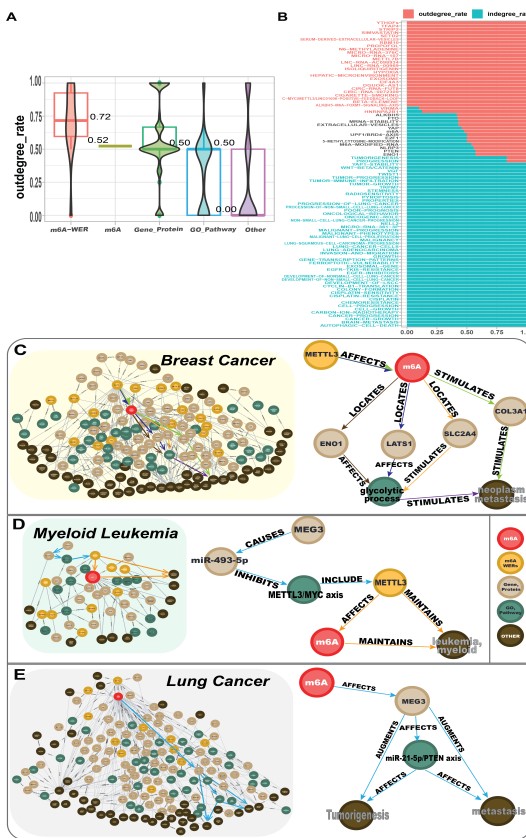

Fig. 3. (A) Outdegree rate of nodes in different categories. (B) Nodes in lung cancer KG with high/medium/low out-degree rates. Cancer-type specific KG of (C) breast cancer, (D) myeloid leukemia, and (E) lung cancer. Extracted pathways are shown to the left. Edge colors are associated with the supporting titles.

### D. m⁶A-KG, a context-aware-KG of m⁶A functions

Next, we demonstrated the utility of reguloGPT to create a KG of the m⁶A-associated MRPs. m⁶A is one of the most predominant mRNA modifications in mammalian cells, present in over 40% of transcripts [29]. The dynamic m⁶A regulation involves various RNA binding proteins (RPBs) including writers (METTL3 & METTL14), which add methyl groups, erasers (ALKBH5 & FTO) which remove it, and readers, (e.g., YTH proteins), which bind to m⁶A sites to decode the regulatory signals to mediate gene expression. It achieves this by regulating stability, alternative splicing, nuclear export, and translation efficiency of mRNA. Additionally, it significantly influences cancer development and progression by modulating mRNA stability and splicing [30]. Despite growing interest, the roles of m⁶A and its writers, erasers, and readers in cancer through gene expression alterations are not fully understood.

*1) Construction of m⁶A-KG with reguloGPT:* We applied reguloGPT to 969 titles without annotations, resulting in the extraction of context-aware relational graphs that depict functions related to m⁶A in diverse contexts. After normalizing the nodes, edges, and contexts, we integrated these relation graphs and the annotated graphs in the benchmark dataset into a comprehensive m⁶A knowledge graph (m⁶A-KG), denoting

molecular regulatory pathways linked to m⁶A. The constructed m⁶A-KG comprises 2442 nodes, 4678 links with 200 edge predicates, and 376 contexts where each link encompassing an average of 0.91 contexts. The average node degree, calculated by aggregating incoming and outgoing edges, is 2.15 with 96% of nodes having less than five degrees and only eight nodes possessing more than 20 degrees. As expected, the m⁶A node emerged as the most connected, with a degree of 65, highlighting its centrality in the network. The top nodes by degree also include key m⁶A writers like METTL3 (degree 47) and METTL14 (degree 23), erasers such as ALKBH5 (degree 24) and FTO (degree 38), and readers like YTHDF1 (degree 28) and YTHDF2 (degree 22). These high degree nodes underscore their vital roles in the regulatory functions of m⁶A. Additionally, nodes like APOPTOSIS (degree 13), GLYCOLYSIS (degree 11), and METASTASIS (degree 10) also exhibited relatively high degrees, indicating m⁶A's associations with these tumor-related entities.

*2) The structure of m⁶A-KG reflects the architecture of molecular regulatory pathways.:* Out-degree rates across each node category revealed a hierarchical structure aligned with that of a molecular pathway (Fig. 3A). Specifically, **m⁶A-WER** and m⁶A have median out-degree rates of 0.72 and 0.52, respectively, suggesting that they occupy upstream positions and reaffirming their roles as key regulators. **Gene_Protein** and **GO_Pathway**, with a median out-degree rate of 0.5, serve as intermediate nodes bridging the upstream regulators with the downstream nodes. Interestingly, Other nodes exhibited a median out-degree rate of 0.00, indicating their positions at the extreme end of the pathway. Close inspection revealed that nodes with high out-degree rates include chemicals or environmental stimuli like simvastatin, propofol, hypoxia, hepatic microenvironment, and cigarette smoking, which are expected to be upstream in pathways (Fig. 3B). Conversely, nodes with high in-degree rates were disease phenotypes or outcomes such as tumorigenesis, tumor growth, cancer progression, radiosensitivity, drug resistance/sensitivity, chemoresistance, or autophagic cell death, which naturally sit at the bottom of pathways (Fig. 3B). The emergent structure of the m⁶A-KG, featuring various stimuli at the top followed by clear upstream m⁶A regulators and downstream phenotype outcomes, exhibits the hallmarks of an MRP.

*3) The m⁶A-KG reveals distinct mechanisms of m⁶A functions across various cancer types.:* We next investigated m⁶A's role in various cancers by leveraging m⁶A-KG's integration of contexts and PMIDs into edges. This feature enabled us to dissect functions specific to certain cancers and to identify those common across multiple types. The m⁶A-KG's contexts included 67 cancer contexts represented by 2558 links defined by 112 edge predicates. Especially, the link 'METTL3 – [*AFFECTS*] – m⁶A' was associated with 35 cancer contexts including 12 TCGA cancer types, signifying METTL3's ubiquitous influence in various cancers and regulatory functions. Moreover, four relations ('ALKBH5 – [*AFFECTS*] – m⁶A,' 'YTHDF2 – [*INTERACTS_WITH*] – m⁶A,' 'METTL3 – [*STIMULATES*] – PROGRESSION,' and

'WTAP – [*AFFECTS*] – m⁶A') were associated with more than 10 cancer types, highlighting the central role of the writer METTL3, the eraser ALKBH5, and the reader YTHDF2 in multiple cancers.

To gain insights into cancer-specific m⁶A-mediated functions, we extracted cancer-specific sub-KGs for breast cancer, leukemia, and lung cancer (Fig. 3C, D, and E). These sub-KGs presented clear hierarchies of MRPs, with m⁶A regulators at the top and disease phenotype nodes downstream. METTL3's widespread association across cancers prompted further examination of pathways centering on this regulator. We focused on pathways supported by edges spanning multiple titles because they could reveal novel functions. The breast cancer sub-KG (Fig. 3C) delineates a complex dual-pathway mechanism, with evidence from five titles [26], [31]–[34], suggesting METTL3's involvement in tumor metastasis through two distinct routes: regulation of COL3A1, crucial for extracellular matrix structure, and alteration of cancer cell metabolism via the glycolytic pathway. This duality suggests that therapeutic targeting METTL3 could simultaneously disrupt key structural and metabolic routes essential to cancer metastasis, offering a promising avenue for multifaceted therapeutic intervention. Moreover, cancer-dependent regulations of MEG3, a tumor suppressor gene, were revealed in lung and leukemia sub-KGs. The leukemia sub-KG (Fig. 3D) suggests that MEG3 modulates miR-493-5p to suppress myeloid leukemia by inhibiting METTL3-mediated m⁶A methylation [35], [36]. Conversely, in lung cancer sub-KG (Fig. 3E), METTL3 methylates MEG3, which facilitates tumorigenesis and metastasis [37]. These distinct regulatory mechanisms were corroborated through a detailed examination of the literature associated with the extracted pathways [38], validating the m⁶A-KG's utility in uncovering new functions.

## V. Conclusion

In this study, we developed reguloGPT, a novel GPT-4 application for the end-to-end KG construction for MRPs. We devised ICL prompting to extract context-aware relational graphs for MRPs. We annotated 400 titles on m⁶A methylation, encompassing various regulations applicable to broader studies. We benchmarked reguloGPT, showing significant performance improvements over existing algorithms, including Mixtral-8x22B and Llama3-70B. Also, we found a strong similarity between manual evaluation and our proposed annotation-free G-Eval evaluation. reguloGPT successfully built a detailed m⁶A-KG with 2442 nodes, 4678 links with 200 predicates, providing a rich map of m⁶A regulatory functions by featuring its unique context-aware edges with associated contexts and PubMed IDs. This design not only allowed us to understand context-specific regulations but also enhances traceability and data verification. The m⁶A-KG revealed distinct mechanisms of m⁶A functions across various cancer types, facilitating a deeper understanding of m⁶A's role in cancer and opening avenues for targeted cancer research and therapy development. The hierarchical structure of the m⁶A-KG mirrors the architecture of MRPs, offering a more intuitive understanding of the

complex interactions and roles within these pathways. Future studies will explore a more systematic G-Eval assessment and relationship extraction, along with improved normalization schemes for edges and contexts. A systematic and effective approach to elucidate novel regulatory functions from the KG will be further developed.

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
