# OpenReview forum: "reguloGPT: Harnessing GPT for End-to-End Knowledge Graph Construction of Molecular Regulatory Pathways"
_IEEE.org/EMBS/BHI/2024/Conference — IEEE BHI'24_

### Official Review · Reviewer_fXHV · 2024-08-07
**ReguloGPT and Its Application in Constructing Knowledge Graphs for Targeted Cancer Research**

**Overall Rating:** 7
**Confidence:** 4

**Other Quality Metrics:**

(a) Clarity of writing: good
(b) Clinical Significance: great
(c) Methodological Novelty: great
(d) Experiments and Results: good

**Questions For The Authors:**

What is the significance of mining knowledge graphs from biomedical literature? Could you elaborate on this?
I noticed some relevant explanations in Result.D: 'm6A-KG, a context-aware KG of m6A functions.' Please consider providing a more detailed summary and highlighting these aspects in the introduction

**Strengths:**

The paper presents an interesting idea and a novel method, characterized by clear research logic and rigorous database construction. The innovative approach to integrating trendy methodologies to advance their research field, coupled with the solid work in building a robust database, constitutes the most significant and enduring contributions of this study.

**Summary Of The Paper:**

The authors developed reguloGPT, a novel GPT-4 application that implements instruction-following conversational (ICL) prompting to generate context-specific relational graphs detailing MRP interactions. They built a benchmark dataset of 400 m6A methylation titles and used it to evaluate and prove the efficacy of the proposed reguloGPT. The authors also proposed an annotation-free evaluation method, G-Eval, and demonstrated its consistency with benchmark evaluations. Finally, they successfully applied reguloGPT to create a comprehensive and detailed knowledge graph, m6A-KG, which maps diverse regulatory mechanisms across cancer types, enhancing our understanding of m6A's role in cancer and supporting targeted research and therapy development.

**Weaknesses:**

The prompts are based on GPT-4 and other cutting-edge large language models (LLMs), which, although state-of-the-art at the time of drafting, may quickly become outdated due to rapid model iterations.

---

### Official Review · Reviewer_EHgX · 2024-08-08
**Reviews of Submission361**

**Overall Rating:** 7
**Confidence:** 3

**Other Quality Metrics:**

(a) Clarity of writing --- good
(b) Clinical Significance --- good
(c) Methodological Novelty  --- good
(d) Experiments and Results --- good

**Questions For The Authors:**

See the weakness.

**Strengths:**

1. The application of GPT-4 for end-to-end KG construction in the domain of molecular biology is innovative. The use of ICL prompts to extract detailed, context-aware regulatory graphs is contemporary.
2. The introduction of context-aware relational graphs that incorporate context directly within the relational edges is a good approach that enhances the semantic accuracy and relevance of the extracted relationships.

**Summary Of The Paper:**

This paper introduces an approach leveraging GPT-4 for the automated construction of Knowledge Graphs (KGs) for Molecular Regulatory Pathways (MRPs). The tool, reguloGPT, uses in-context learning (ICL) to generate context-aware relational graphs from textual descriptions of MRPs, particularly focusing on m6A mRNA methylation. This approach addresses the limitations of current methods in capturing complex, hierarchical relationships and contextual nuances within MRPs. The paper details the creation of a benchmark dataset from 400 annotated PubMed titles and evaluates reguloGPT's performance in extracting accurate and contextually relevant information compared to existing algorithms and other large language models (LLMs). The study also explores the application of reguloGPT to a larger dataset, resulting in a comprehensive m6A-KG that elucidates m6A's regulatory roles across various cancers.

**Weaknesses:**

1. The effectiveness of reguloGPT is likely contingent on the quality and specificity of the input data. The paper does not sufficiently explore how data quality or variations in textual descriptions might influence the output KGs. Additional ablation studies on this aspect would be helpful.
2. Could you provide further details on the design of the prompt?
3. The experimental design as presented is deterministic. It may be advantageous to adopt a probabilistic approach, such as updating experimental designs based on specific criteria using Bayesian methods or incorporating active learning to expedite the experimentation process.

---

### Official Review · Reviewer_9ZZB · 2024-08-11
**reguloGPT: Harnessing GPT for End-to-End Knowledge Graph Construction of Molecular Regulatory Pathways**

**Overall Rating:** 7
**Confidence:** 3

**Other Quality Metrics:**

(a) Clarity of writing: great
(b) Clinical Significance: good
(c) Methodological Novelty: great
(d) Experiments and Results: good

**Questions For The Authors:**

Why the four established algorithms were selected for baselines? The criteria used for the selection should be discussed
How were the known problems of LLMs (hallucinations, etc) addressed? Or at least, how do they impact the results? This should also be discussed
There is a huge amount of works using LLMs, some disuussion about this is needed.

**Strengths:**

Well written paper
Relevant topic in an active domain
Strong validation

**Summary Of The Paper:**

This paper proposes the use of ChatGPT to extract knowledge graphs about molecular regulatory pathways. To do that, the authors first collected a dataset of paper titles containing regulatory pathways for m6A. Some of this titles were used by experts to generate knowledge graphs that are then used to compare with the results provided by ChatGPT. Additionally, the chatGPT was also used to automatically evaluate the generated knowledge graphs and compared with the other LLMs

**Weaknesses:**

The text on most figures is too small to be read
There is a lot of works using ChatGPT, comparison or at least discussion about related or similar works is needed

---

### Decision · Program_Chairs · 2024-09-23

Accept